# C5 methylation confers accessibility, stability and selectivity to picrotoxinin

Guanghu Tong [1], Samantha Griffin[2], Avery Sader[2], Anna B. Crowell[3], Ken Beavers[2], Jerry Watson[2], Zachary Buchan [2], Shuming Chen[3] ✉ & Ryan A. Shenvi [1] ✉

Minor changes to complex structures can exert major influences on synthesis strategy and functional properties. Here we explore two parallel series of picrotoxinin (PXN, **1**) analogs and identify leads with selectivity between mammalian and insect ion channels. These are the first SAR studies of PXN despite its >100-year history and are made possible by advances in total synthesis. We observe a remarkable stabilizing effect of a C5 methyl, which completely blocks C15 alcoholysis via destabilization of an intermediate twist-boat conformer; suppression of this secondary hydrolysis pathway increases half-life in plasma. C5 methylation also decreases potency against vertebrate ion channels (γ-Aminobutyric acid type A (GABA$_A$) receptors) but maintains or increases antagonism of homologous invertebrate GABA-gated chloride channels (resistance to dieldrin (RDL) receptors). Optimal 5MePXN analogs appear to change the PXN binding pose within GABA$_A$Rs by disruption of a hydrogen bond network. These discoveries were made possible by the lower synthetic burden of 5MePXN (**2**) and were illuminated by the parallel analog series, which allowed characterization of the role of the synthetically simplifying C5 methyl in channel selectivity. These are the first SAR studies to identify changes to PXN that increase the GABA$_A$-RDL selectivity index.

Natural product total synthesis allows analysis of structure-activity relationships (SAR) when a robust pathway is established[1]. These efforts can enable gain of function through deep-seated modification of structure[1], but synthetic difficulty can hinder optimization campaigns. To simplify material access, either the complexity of the natural product target (TGT) can be lowered while maintaining receptor contacts[2] or new strategies and methods can improve efficiency[2]. In the former case, large simplifications to the structure can affect physicochemical properties, whereas in the latter case the structural liabilities of the natural product are maintained. One alternative is to address structural liabilities in the design stage and identify novel analogs with enhanced functions[3,4] that maintain most physicochemical properties of the natural product[3]. Because structural changes introduce alternative retrosynthetic disconnections, this approach may also improve material access[3].

Here we explore the modification, stabilization, and activity of picrotoxinin, a potent GABA$_A$ receptor antagonist. Appendage of a methyl at C5 enhances synthetic access, stabilizes the scaffold against solvolysis and increases selectivity between invertebrate and vertebrate ion channels (Fig. 1). These are the first studies to improve the selectivity index of PXN among ligand-gated ion channels (LGICs) and stabilize PXN towards chemical and metabolic degradation[5].

Seeds of *Anamerta cocculus* have for centuries featured as poisons[6,7] due to their content of picrotoxinin[5], a sesquiterpene that rapidly absorbs, penetrates the mammalian brain, and potently antagonizes GABA$_A$ receptors[8–10]. PXN has become a widespread tool

[1]Department of Chemistry, Scripps Research, 10550 North Torrey Pines Road, La Jolla, California 92037, USA. [2]Corteva Agriscience, 9330 Zionsville Road, Indianapolis, Indiana 46268, USA. [3]Department of Chemistry and Biochemistry, Oberlin College, 119 Woodland Street, Oberlin, Ohio 44074, USA. ✉e-mail: shuming.chen@oberlin.edu; rshenvi@scripps.edu

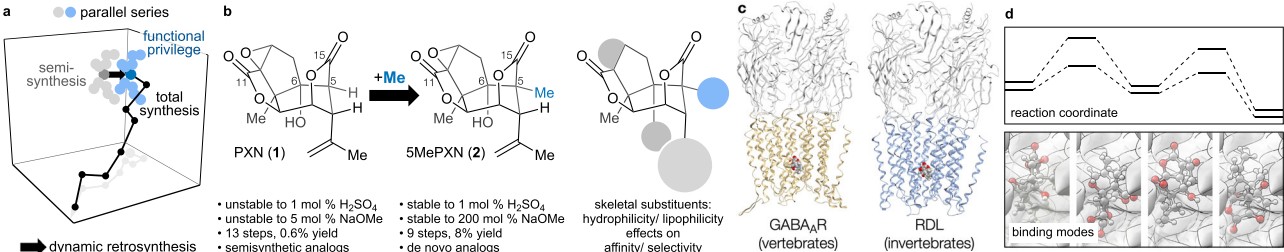

**Fig. 1 | Overview: target modification to explore functionally privileged chemical space. a** Chemical space plot of parallel series to explore the effects of a scaffold uniquely accessible through total synthesis; (**b**) C5 methylation increases stability to base and acid, increases yield, decreases required steps and increases receptor selectivity; (**c**) Assay against GABA_A and RDL receptors, representative of vertebrate (e.g. human) and invertebrate (e.g. insect) ligand-gated ion channels (LGICs), respectively. Left: rat GABA_A homology model from PDB 6×40 template with sequence from *R. norvegicus*, gold. Right: fly RDL homology model from PDB 6×40 template with sequence from *D. melanogaster*, blue. **d** Computational analyses provide models for increased stability and selectivity of the 5MePXN series. PXN picrotoxinin, 5MePXN 5-methylpicrotoxinin.

in neuroscience to identify and/or ablate GABAergic signaling[11,12]. Mammalian GABA_A receptors are inhibitory LGICs, which form an extensive phylogenetic tree of membrane-bound pentameric protein complexes[13]. Despite the widespread availability of PXN, no semi-synthetic analogs have demonstrated improved properties: selectivity among ion channels or stability to degradation[5]. PXN is surprisingly unstable. Rapid and irreversible solvolysis occurs with 5 mol % sodium methoxide in methanol[14] or 1% sulfuric acid in water[15]. Reversible solvolysis occurs in pH 7–9 water[16], or mouse plasma (half-life ~ 1 h, vide infra)[17]. In each case, the degradation products are weakly active or inactive in vivo[9,16,17]. Total synthesis might solve these problems with skeletal changes that stabilize the scaffold and allow diversification. However, no syntheses have delivered analogs for assay against LGICs, selectivity among receptors or stability to solvolysis. Members of the LGIC phylogenetic tree represent important targets in infectious disease and agriculture as components of invertebrate nervous systems[11–13]. The ability to modulate the vertebrate/invertebrate selectivity of ion channel antagonists dictates potential use[18,19]. For example, the long-discontinued insecticide dieldrin persists in soil, intensifies up the food chain due to lipophilicity and potently antagonizes both mammalian GABA_A and insect RDL LGICs, representing a threat to food supplies[20].

Recently, we reported a short synthesis of PXN that identified a retrosynthetic methylation transform to control stereochemistry and conformation of early intermediates[21]. Whereas excision of this extra methyl required multiple steps, retention of the methyl allowed a more rapid and higher-yielding entry into PXN chemical space (PXN: 13 steps, 0.6% overall vs. 5MePXN: 9 steps, 8% overall). 5MePXN retained binding to rat GABA_A receptors and we wondered whether the synthesis might provide analogs that exhibit selectivity among LGICs. Here we show that the total synthesis of 5MePXN allows, for the first time, diversification of the PXN scaffold, optimization of selectivity between mammalian GABA_A and insect RDL receptors, and a stabilization of the PXN scaffold to both strong acid and strong base. We probe the effects of the C5Me by constructing parallel series of PXN and 5MePXN analogs through total synthesis and semi-synthesis, respectively (Fig. 1a), to illustrate how C5 methylation significantly changes stability, accessibility (Fig. 1b), and receptor selectivity (Fig. 1c). The basis for stabilization and receptor selectivity is then probed computationally (Fig. 1d).

## Results

### The design and synthesis of picrotoxinin analogs
The 5 rings, 8 stereocenters, and 6 oxygens of PXN comprise a Böttcher complexity ($C_m$)[22,23] of 467.61 mcbits packaged into 228 Å³, to give a density of 2.05 mcbits/Å³. Its small volume allows PXN to fit into the narrow space near the desensitization gate[24] of GABA_A receptors[24] (largest pore radius *ca.* 3 Å in α1β3γ2L)[25]. Methylation of C5 increases

the $C_m$ to 480.1 (+ 3%), the ligand volume by 14 Å³ (+ 6%), maintains the information density (1.98 mcbits/Å³) and appears to access a hydrophobic pocket defined by residues L259/ T256 (β3 subunit) and T261 (α1 subunit)[25,26]. In a preliminary screen, we found that 5MePXN competed with [³H]-*t*-butylbicycloorthobenzoate ([³H]TBOB) for binding to rat cerebral cortex with an IC_{50} of 2.1 ± 0.3 µM[21], corresponding to a 10-fold loss in potency from PXN. We wondered if this potency loss at mammalian receptors could serve as an advantage if the 5Me series bound invertebrate receptors with greater affinity, broadening the selectivity index of PXN analogs between mammals and insects.

The high oxidation state, lack of modifiable functional groups and complex structure of PXN obstructed functionalization in prior studies[5]. Moreover, interactions between PXN and GABA_A receptor in the narrow pore do not allow expansive derivatization of the PXN scaffold: e.g. acylation of the C6 hydroxyl or C15 bridge lactone opening ablates binding[5]. Thus, we pursued analogs based on previous computational[18] and experimental[25,26] models that 1) minimized volume and conformational changes using single atom replacements[1], 2) probed lipophilic modification at C12 to avoid the reported inactivity of picrotin (PTN, **10**) at RDL[27], and 3) explored hydrogen bond network disruption (de-epoxidation, hydroxylation or fluorination[27]; Fig. 1b).

The divergent syntheses of parallel PXN and 5MePXN analogs are shown in Fig. 2. Diversification of the 5Me series began from late-stage intermediate **7**, 2.7 grams of which were accessed in a single-pass scale up over 6 steps (see **3** to **7**) from dimethylcarvone (**3**) in 27% overall yield. Oxidation of the *gem*-dimethyl motif to the bridging lactone could be carried out in one step (Pb(OAc)_4, I_2, 31%) or two steps (AgOAc, I_2, 51%; RuCl_3, NaIO_4, 79%). The formation of 5MePTN (**11**) was accomplished in excellent yield (91%) according to our previous synthesis of **10** using Mukaiyama hydration[21,28,29]. 12FPXN (**12**) and 12F5MePXN (**13**) were synthesized through the highly effective Boger hydrofluorination (35% and 40% yield, respectively)[30]. The C12/ C13 cyclopropane was introduced on the picrotoxinin scaffold through Simmons–Smith cyclopropanation (CH_2I_2, Et_2Zn) to furnish CypPXN (**14**) in modest yield (52%). Under the same conditions, 5MePXN yielded the desired product in less than 5% yield, but an alternative procedure was developed using CH_2ICl[31], which resulted in a 1:3 mixture of Cyp5MePXN (**15**) and 8,9-deoxy-cyclopropane-5MePXN (Supplementary Information). Fortunately, treatment of this mixture with freshly prepared dimethyldioxane (DMDO) converted 8,9-deoxy-cyclopropane-5MePXN to **15** in 72% yield (two steps). Cyclopropanation at C12/ 13 not only delivered **14** and **15** but also allowed us to prepare C4 *tert*-butyl analogs. Whereas verified GABA_AR antagonists like bilobalide, TBPS (*t*-butylbicyclophosphoro-thionate) and TBOB contain a *tert*-butyl group[32], analogous PXN substitution has not been assayed (the near-isosteric tertiary alcohol PTN (**10**) significantly reduces binding

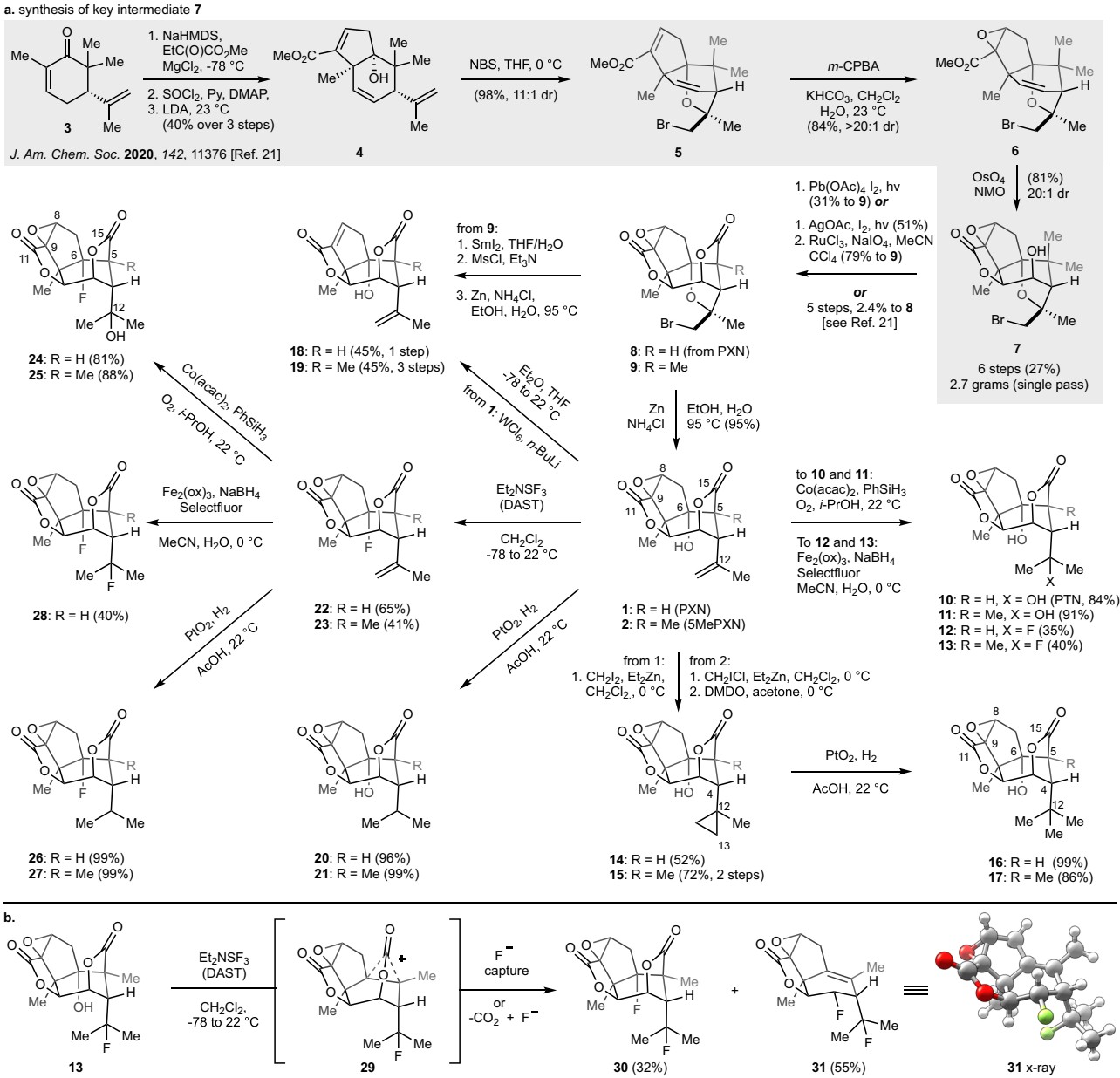

**Fig. 2 | Twenty-one analogs prepared for biological testing via parallel diversification of PXN and 5MePXN. a** Analogs probe oxygenation and fluorination patterns; (**b**) C5-methylation affects reactivity in unexpected ways (see also Fig. 3).

affinity). Two hydromethylation methods were attempted on **1** and **2**[33,34], but only hydrogenation of corresponding cyclopropane (**14** and **15**) with 50 mol% Adams' catalyst (PtO₂) yielded 12MePXN (**16**) and 12Me5MePXN (**17**) in 99% and 86% yield, respectively. Degradation product 8,9doPXN (**18**) was accessed according to Trost's method (WCl₆, *n*-BuLi, 45% yield)[35]. The corresponding 5Me analog was accessed by exposure of 5MeBrPXN (**9**) to samarium diiodide (SmI₂) in degassed THF/H₂O (20:1, v:v), which led to chemoselective epoxide opening without reducing the primary bromide. Elimination (MsCl, Et₃N) yielded 8,9do5MeBrPXN, which was then heated under reductive debromination conditions (Zn, NH₄Cl, 95 °C) to deliver 8,9do5MePXN (**19**) in 45% yield over three steps. HPXN (**20**), another natural GABAₐR antagonist, could be conveniently obtained from PXN through hydrogenation (PtO₂, 96% yield); H5MePXN (**21**) was prepared in a similar manner in quantitative yield.

Hydration of the Δ¹²,¹³-alkene (PXN to PTN) dramatically reduced binding affinity to GABAₐ receptors[5] but not RDL (see below). We wondered whether replacement of one or both C12 and C6 hydroxy groups with fluorine might disrupt internal hydrogen bonding and, combined with C5 methylation, differentially affect affinity for receptors. Advanced synthetic intermediate 6FPXN (**22**) was prepared from **1** (65% yield) by reaction with diethylaminosulfur trifluoride (DAST)[27,36], and further employed as starting material for the synthesis of analogues 6FPTN (**24**) and 6FHPXN (**26**). In parallel, 6F5MePXN (**23**) was synthesized in 41% yield. Cobalt catalyzed Δ¹²,¹³-alkene hydration smoothly proceeded on 6FPXN and 6F5MePXN, delivering **24** and 6F5MePTN (**25**) in 81% and 88% yield, respectively. Dihydro-congeners 6FHPXN (**26**) and 6FH5MePXN (**27**) were prepared as above using PtO₂-catalyzed hydrogenation.

6F12FPXN (**28**) arose via Boger hydrofluorination (Fe₂(ox)₃, Selectfluor, NaBH₄) in 40% yield (accompanying alkene reduction byproduct **26**) but confounded assignment due to an unexpected, strong fluorine-fluorine coupling by ¹⁹F NMR ($J_{F,F} = 31.2$ Hz)[37]. Correlation to the 5Me series was helpful: treatment of **13** with DAST at low

temperature followed by weak basic workup yielded two crystalline compounds after silica chromatography in 55% and 32% yield (Fig. 2b). Single-crystal X-ray diffraction of both structures identified the major product as **31**, an unexpected fragmentation product, and the minor product as 6F12F5MePXN (**30**). The [19]F coupling constant of **30** was found to be 27.8 Hz, aiding the assignment of **28** via the relationship between C6-F and C12-F. The additional C5-methyl must significantly stabilize the high-energy, non-classical cation **29**, leading to the unexpected reactivity of decarboxylation and fluoride capture at C3 or C6. The unexpected stabilizing role of the C5-methyl heralded more remarkable effects in classic solvolytic degradations of PXN.

Picrotoxinin undergoes reversible hydrolysis to lose efficacy upon storage in water between pH 7–9 or in mouse plasma[17]. Whereas the site of initial, reversible hydrolysis had long been hypothesized to be C15, we recently observed by [1]H NMR that the actual site of hydrolysis with 1 equiv. NaOH was C11[38]. Hydrolysis at C11 is reversible if only one equivalent of hydroxide is used: protonation with HCl returns PXN with only small amounts of degradation product (picrotoxic acid, see **32**, Fig. 3a) derived from C15 hydrolysis[38]. In contrast to C11, hydrolysis at C15 is irreversible − a point-of-no-return due to concomitant epoxide opening or cyclohexane relaxation, with no precedent for chemical or biochemical reversion: exposure to excess base led to complex decomposition[14,39–42]. We had previously observed 5-methyl substitution to decrease degradation by halving the pseudo-first order rate constant ($t_{1/2} = 60$ vs. 120 h, $D_2O$, 8.2 pH*)[43], a result of fast reversible and slow irreversible degradation. We had not probed the effect of C5 substitution on irreversible degradation of PXN with strong acid, alkoxide base, or incubation in plasma. These studies are important because chemical manipulations of PXN to generate analogs (e.g. Fig. 2) must otherwise avoid acid and base due to instability[5], and plasma stability influences future biological application to human disease.

Warming PXN with classical conditions[5] to promote irreversible degradation (1% $H_2SO_4$; 0.18 M, aqueous) led to clean conversion to picrotoxic acid (**32**) in 24 h via cleavage of the C15 lactone, cyclohexane ring-opening and intramolecular epoxide addition (Fig. 3a, top)[5]. 5MePXN, in contrast, did not react at all under identical conditions! Addition of 5 mol % NaOMe to a solution of PXN (**1**) in $d_4$-methanol at 22 °C led to an equimolar mixture of $d_5$-methyl picrotoxate ($d_5$-**33**) and $d_9$-dimethyl picrotoxinin dicarboxylate ($d_9$-**34**) in 6 h (Fig. 3a, bottom). In contrast, we did not observe any consumption of 5MePXN under the same conditions (Fig. 3b). To ensure an NMR-inactive acidic impurity was not consuming the catalytic strong base, we prepared a 1:1 mixture of PXN and 5MePXN and added 5 mol% NaOMe so that both substrates were exposed to identical conditions: PXN completely degraded, whereas 5MePXN completely persisted (Fig. 3c and d). Even addition of 2 equivalents NaOMe had no effect on 5MePXN (Fig. 3b and SI), far exceeding the tolerance of PXN. Thus, C5 methylation stabilized PXN against buffered water hydrolysis, acidic hydrolysis and alkaline methanolysis (Fig. 3a–d, see also ref. 21). This profound stabilization highlighted the unique effects to reactivity imparted by C5 methylation. Furthermore, it opened a window into understanding differences in blood degradation of 5MePXN.

We investigated the relative rates of esterase-mediated hydrolysis[17] observed in mouse and human plasma, to evaluate potential for future translational applications. C5 methylation led to a doubling (2.1X) and near tripling (2.7X) of the extrapolated half-life in mouse ($t_{1/2} = 97$ vs. 207 min) and human plasma ($t_{1/2} = 108$ vs. >289 min), respectively (Fig. 3e). Due to the high stability of the C15 lactone, identified above, combined with the kinetic lability and reversible opening of the C11 lactone identified previously[38], enzymatic degradation in serum likely reflects a combination of reversible C11 hydrolysis[38] (unaffected by C5 methylation) and irreversible C15 hydrolysis (obstructed by C5 methylation)[17]. No prior studies have identified PXN modifications that enhance metabolic stability, partly due to the difficulty of synthetic modification and partly due to the prior incorrect assignment of the site of kinetic hydrolysis (C11 vs. C15)[38].

## Computational models for stability and affinity

The structural basis for resistance to solvolysis was probed with density functional theory (DFT) computational models to weigh three competing hypotheses: 1) a high barrier to nucleophile addition due to C5Me obstruction of Bürgi-Dunitz[44] or Heathcock's Flippin-Lodge[45,46] angles; 2) a high barrier to conformational change in methanolysis intermediates; or 3) a high barrier to subsequent reaction of the initial methanolysis product (i.e. formation of 5Me-**33** or 5Me-**34**). Structures and free energies of transition states and intermediates for base-catalyzed methanolysis were calculated at the ωB97X-D/def2-TZVPPD, SMD ($H_2O$)// ωB97X-D/def2-SVP, SMD ($H_2O$) level of theory in Gaussian 16, which has been shown to perform well in the modeling of transformations involving anionic intermediates[47,48]. Comparison of relative energies of intermediates and transition states of PXN (**1**) and 5MePXN (**2**) solvolyses suggested that the resistance of **2** to methanolysis was unlikely to be caused by kinetic factors (hypotheses #1 and #2) and more likely due to thermodynamic factors (Fig. 3f, g). Whereas the C15 methanolysis of **1** to anion **B** could be reversed, epoxide opening (to **C**) and C11 lactone opening (to **D**) provide competitive escape pathways for **B** (14.1 kcal/mol barrier to product, 12.8 kcal/mol barrier to starting material). In contrast, calculated energies of the analogous C5Me intermediates reside on an energetic plateau above **2**; reversion to **2** encounters only a 5.9 kcal/mol barrier, whereas the escape pathway to epoxide-opened product **H** requires 12.5 kcal/mol and an overall 19.7 kcal/mol rate-determining step. The 7.4 kcal/mol difference between the intramolecular epoxide opening barriers for picrotoxinin and 5-Me-picrotoxinin roughly translates to a $1.6 \times 10^5$-fold difference in rate, in good agreement with the experimental differences in rate. The major steric interactions that raise the energies of **TS-3** and **TS-4** compared to their picrotoxinin counterparts are shown in Fig. 4. For the nucleophilic attack TS (**TS-3**), the 5-Me substituent clashes directly with the incoming oxygen nucleophile (H...O distance 2.34 Å, or 86% of the sum of the van der Waals radii). For both the C15 lactone-opened intermediates and the intramolecular epoxide opening transition states (**TS-4**), the free energies are raised due to the steric repulsions caused by the pseudoaxial 5-Me group across the cyclohexane ring. Return to the C15 lactone alleviates these repulsive interactions.

## Receptor selectivity

The structurally privileged 5-methyl series was assayed for binding to both GABA$_A$ receptors (rat cortex membrane preparations) and RDL (expressed in *Xenopus* oocytes) and compared directly to the parent PXN (C5-H) series. The parallel series of PXN and 5MePXN analogs allowed us to interrogate the effect of the synthetically-enabling and stabilizing C5 methyl.

The PXN-series (Fig. 5a) is plotted in descending order of rat cortex IC$_{50}$ values and correlated with arrows to the corresponding 5Me analogs. In this way, the effect of C5 methylation can be visualized easily, along with other trends. Hydrophobic substitution of C12/C13 (-CH$_2$-, -CH$_3$) maintained potency, whereas polar substitution (-OH, -F) reduced potency and alkene hydrogenation lay somewhere in between. Among the more potent compounds, we found that replacement of the C6 hydroxyl with fluoride had little effect on potency as did deoxygenation of the C8,9-epoxide. The least potent analogs included the natural product PTN and its 6F-analog, providing a series that spanned almost 3 orders of magnitude of potency. Across the series, 5MePXN analogs recapitulated the PXN SAR, albeit at lower potencies. More important, however, was the contrast between GABA$_A$ and RDL patterns.

Plotting the 5MePXN series in the same order as Fig. 5a revealed that potencies of PXN analogs at RDL did not follow the same trends.

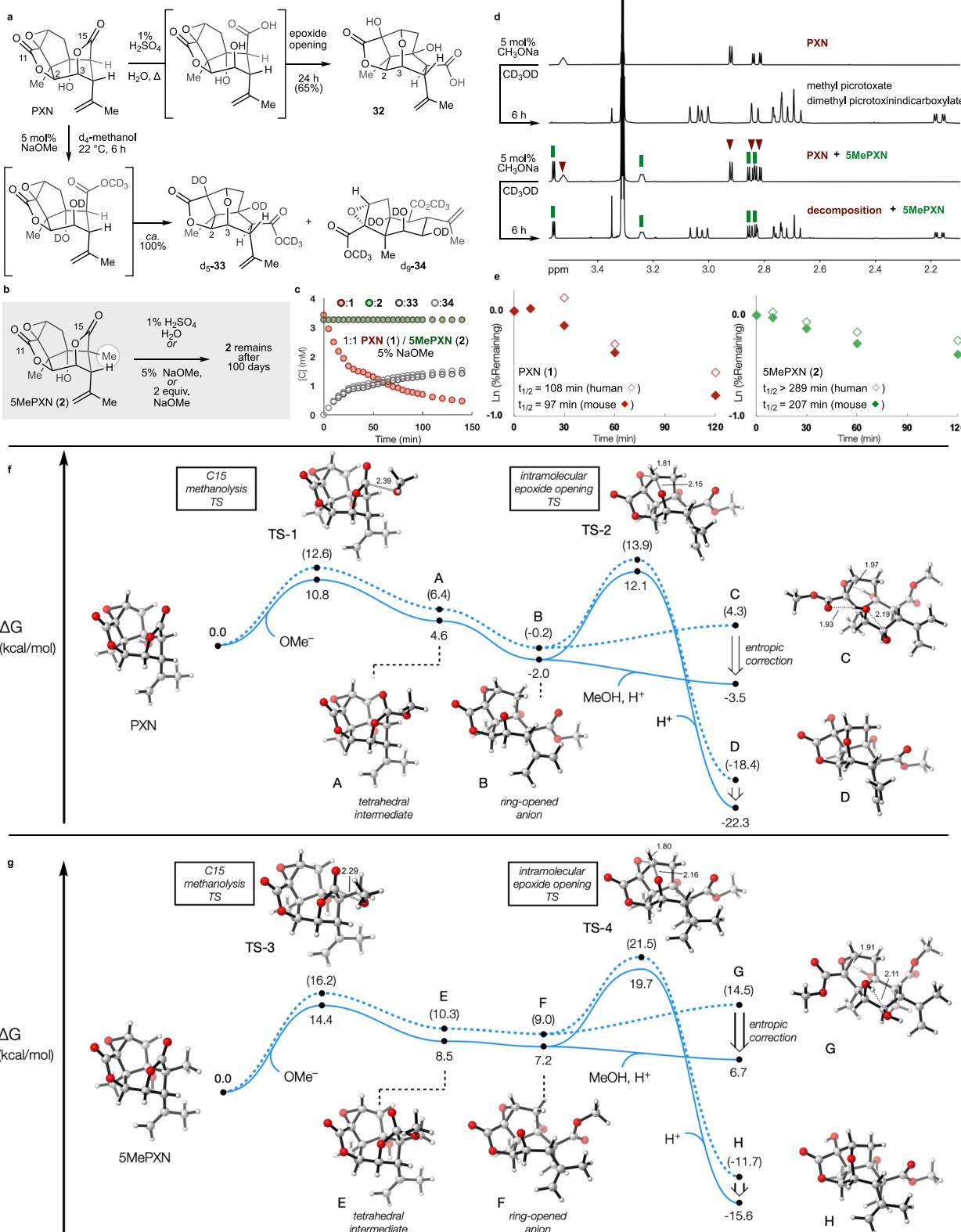

**Fig. 3 | Experimental and computational analysis of stability imparted by C5-methylation.** **a** Degradation of **1** using common acidic and basic conditions; (**b**) Attempted degradation of 5MePXN (**2**) using the same conditions; (**c**) In situ [1]H NMR monitoring of 1:1 PXN/ 5MePXN in 5 mol % NaOMe/ d4-methanol; (**d**) Competition experiments with **1** and **2** using 5 mol % NaOMe in d4-methanol; t = 0 and 6 h, w/ and w/o **2**; (**e**) Degradation is also retarded in mouse and human plasma (n = 2). **f**, **g** Calculated structures and free energies of transition states and intermediates for base-catalyzed methanolysis at the ωB97X-D/def2-TZVPPD, SMD (H2O)// ωB97X-D/def2-SVP, SMD (H2O) level of theory. Interatomic distances are in Å. Dashed curves denote raw calculated values; solid curves denote values with translational entropy corrections accounting for concentration gradients (see Supplementary Information).

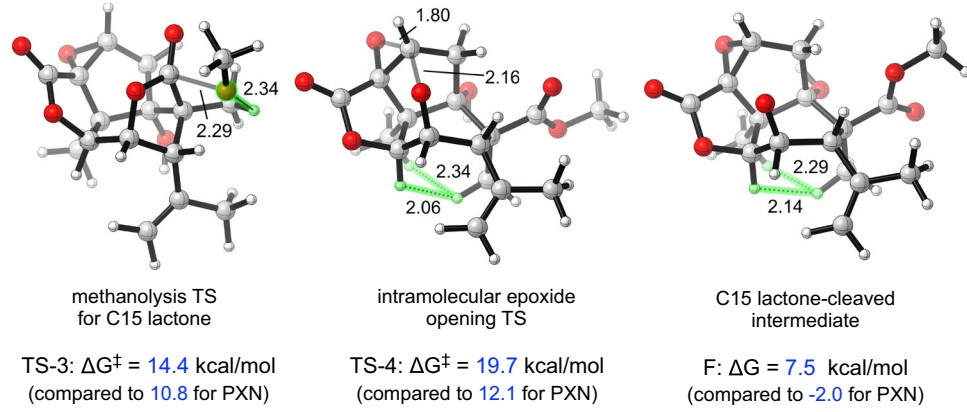

| methanolysis TS<br>for C15 lactone | intramolecular epoxide<br>opening TS | C15 lactone-cleaved<br>intermediate |
|---|---|---|
| TS-3: ΔG‡ = 14.4 kcal/mol<br>(compared to 10.8 for PXN) | TS-4: ΔG‡ = 19.7 kcal/mol<br>(compared to 12.1 for PXN) | F: ΔG = 7.5 kcal/mol<br>(compared to -2.0 for PXN) |

**Fig. 4 | Relevant interactions with C5Me.** Although C5Me clashes with the incoming MeO- nucleophile, the kinetically-relevant effects appear to be diaxial interactions in high-energy twist-boat intermediates.

Instead of mirroring the GABA$_A$R potency trends, C5-methylation increased the RDL potency of 6F and 6F12FPXN, as well as 6FPTN, leading to a ≥ 200X selectivity index between vertebrate and invertebrate receptors and potency at RDL that slightly exceeded that of PXN itself. 6FPXN had been reported previously to lose channel blocking ability[27], but we observed similar or increased potency. Whereas the complex picrotoxane member picrodendrin O has shown high RDL affinity, its selectivity derives from a complex ring at C9 and leads to only a 76-fold selectivity index over GABA$_A$[49]. Its isolation yield of 400 ppb limits its potential[50]. In contrast, selectivity in this 5MePXN series derives from simple core modifications and accompanies high scaffold stability. These are the first assays of PXN analogs generated by total synthesis[35] and the only channel-selective PXN analogs reported to date[51]. We are hopeful that these observations of differential binding may allow 5MePXN and its analogs to be brought to bear on challenging problems in insect crop predation, pest control, and eventually, infectious disease[11,12,19].

**Computational binding models**

To probe the basis for selectivity of **30** for RDL and against GABA$_A$ we constructed in silico models (Fig. 6) using literature coordinates[52]. The helical region of human GABA$_A$ with picrotoxinin bound (PDB 6×40) was used as a template structure to build homology models of GABA$_A$ in *R. norvegicus* and RDL in *D. melanogaster* (see Fig. 5d). Two sequences of the β-unit in *D. melanogaster* RDL (NCBI AAA28559, UniProt P25123) were analyzed for similarity to the template structure. The NCBI sequence was used to build the RDL model due to greater homology with each chain of the template structure. Protein structures were prepared, and protonation issues were fixed using Structure Preparation in MOE (for a full description of computational modeling building and refinement, see the Supplementary Information). Initial binding poses of PXN and **30** were taken from the crystal pose of PXN from the template structure.

Differential selectivity of PXN and **30** in rat GABA$_A$ and fly RDL were assessed with short molecular dynamics (MD) simulations and MM/GBSA calculations of relative binding free energies. Briefly, initial binding poses were minimized with a maximum of 5,000 steps and subjected to 30 ps production runs at 100 K. In our experience, short MD runs at low temperature allow more effective direct comparisons of binding interactions of different ligands in the same active site by decreasing sidechain fluctuations. As a direct consequence, this procedure reduces the large error bars that typically arise from binding free energy calculations. Heavy-atom restraints were imposed on all receptor atoms farther than 8 Å from the ligand. Snapshots for MM/GBSA calculations were extracted every 0.2 ps for a total of 150 conformations of each complex. Analog **30** loses an average of 2 kcal/mol

binding free energy relative to PXN in the rat GABA$_A$ model but gains about 4 kcal/mol in fly RDL relative to PXN, in qualitative agreement with experimentally observed selectivity.

Within the rat GABA$_A$ model, PXN and **30** accept a hydrogen bond from T287 through the epoxide oxygen atom. After MD simulation, PXN displays additional, albeit weak, electrostatic interactions with the sidechain alcohol oxygen atoms of T280 and T309, the latter of which is not observed in the binding pose of **30** (Fig. 5e). Within the fly RDL model, PXN and **30** interact with T294 through the epoxide oxygen atom, analogous to the hydrogen bond seen in the rat GABA$_A$ model. Specific to the fly RDL model are electrostatic interactions with backbone carbonyl oxygen atoms of S286 and A290 and a strong hydrogen bond between the ester carbonyl of the ligand and the sidechain alcohol of T294 on a separate helix (Fig. 5f). These additional interactions shown by MD relaxation and simulations provide a rational explanation for the increased potency of **30** in *D. melanogaster* that is not observed in *R. norvegicus*. While not assessed quantitatively from our simulations, the overall volume of **30** relative to **1** (247 Å$^3$ vs. 229 Å$^3$) may exceed the narrow pore volume of the GABA$_A$R, imparting some degree of differential selectivity among the two organisms. We could not definitively rule out with experimental data the hypothesis from a referee that narrower regions of the GABA$_A$R channel prevent access to the PXN binding site by **30** (as opposed to a hypothesis of low binding affinity to the site itself). However, all analogs antagonized the PXN-site binder ³H-TBOB to varying degrees, suggesting that either access to the PXN binding site is reasonably unobstructed, or PXN analogs allosterically modulated ³H-TBOB binding. This difference in pore sizes is qualitatively reproduced by the transparent surfaces in Fig. 5e, f. The general tolerance of RDL to C5 methylation (Fig. 5c) compared to GABA$_A$ suggests that other LGICs homologous to RDL may be similarly accommodating.

To investigate the potential for ligands to deviate from the active site and diffuse through the channel, docked poses of PXN (**1**) and **30** in the more voluminous fly RDL model were subjected to 1 μs molecular dynamics simulations. More specifically, eight runs of 125 ns with a 0.2 fs timestep and different initial velocities were conducted independently and combined to alleviate the propagation of force field errors in long simulations. The average RMSD values of **1** and **30** are 0.06 ± 0.01 Å and 0.13 ± 0.06 Å, respectively. Interestingly, **30** deviated the most from its initial binding pose and has larger fluctuations throughout the simulation than **1** despite the larger size of the former, with the difference being statistically significant (*t* stat: 274, p « 0.001). Horizontal lines with regular fluctuations in the RMSD plots of **1** (Fig. 7, top) and **30** (Fig. 7, bottom) provide evidence against rotation or translation of the ligands within or away from the active site.

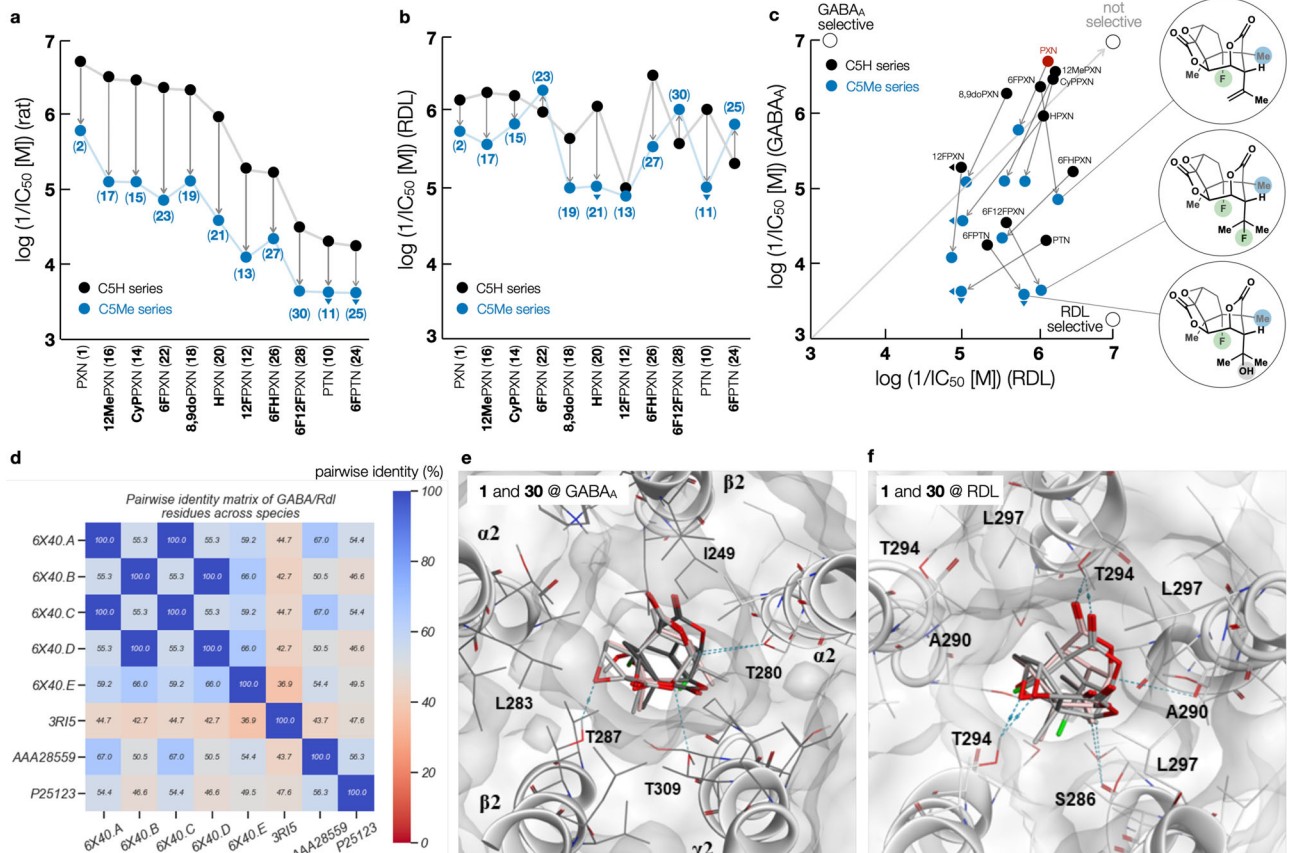

**Fig. 5 | Experimental and computational analysis of LGIC binding and selectivity.** **a** Relative potencies of PXN and 5MePXN analogs at GABA$_A$R ([$^3$H]-TBOB, $n = 1$); (**b**) Relative potencies of PXN and 5MePXN analogs at RDL receptors measured by electrophysiology as a proxy for channel binding ($n = 4$); (**c**) Selectivity between vertebrate (GABA$_A$) and invertebrate (RDL) receptors by PXN analogs: C5-methylated analogs are selective for invertebrate receptors; (**d**) Pairwise identity matrix of GABA/RDL residues across species; (**e**) PXN (**1**) and analog **30** in the rat GABA homology model, showing crystal binding poses of PXN (PDB: 6×40; dark gray), PXN after MD (pink), and **30** (light gray) interacting with T287 (β2), T309 (γ2), and T280 (α2). The binding free energy of **30** is predicted to be 1.8 ± 1.5 kcal/mol above that of **1** at GABA$_A$. Orientation of the binding pocket is kept fixed to allow direct comparison of binding poses; (**f**) PXN (**1**) and analog **30** in the fly RDL homology model. Crystal binding poses of PXN (PDB: 6 × 40; dark gray), PXN after MD (pink), and **30** (light gray) show varying degrees of interaction with the fly RDL homopentamer. After forcefield minimization, the crystal and MD poses of PXN converged. Interactions with S286, A290, and T294 are indicated with dotted lines.

## Discussion

Despite its long history, picrotoxinin's instability, complexity, and toxicity have discouraged its use in agriculture and medicine. Here we show that C5 methylation imparts stability, accessibility, and selectivity. This discovery came from the identification of 5MePXN as a target of convenience during recursive retrosynthetic analyses and synthetic studies toward PXN itself[17], despite the fact that 5MePXN is more complex (C$_m$)[22,23]. This 5MePXN series might be best brought to bear on infectious disease if affinity can be increased towards the structurally related LGIC receptor GluCl, associated with pathogenic protostomes like flatworms and roundworms[20], aided by the experimental data and computational models described here. In addition to the observation that 5MePXN can deliver stabilized, receptor-selective analogs, this study illustrates how target modification at the design stage can deliver important leads for functional exploration[2,53–55].

## Methods

Unless otherwise noted, all experiments were run in flame-dried glassware under an atmosphere of argon. A stir bar is always present in the reaction vessel. Hexane, dichloromethane (DCM), toluene, ethyl acetate (EtOAc), diethyl ether (Et$_2$O), tetrahydrofuran (THF), acetone, dimethylsulfoxide (DMSO), methanol (MeOH), isopropanol (*i*-PrOH), *N*-dimethylformamide (DMF), Acetonitrile (MeCN) and triethylamine (Et$_3$N) were purchased from Sigma Aldrich, Fisher Chemicals or Acros Organics and used without further purification. All anhydrous solvents were purchased from Fisher Chemicals, Sigma Aldrich or Acros Organics and used without further purification, unless otherwise stated. Reactions were monitored by thin layer chromatography (TLC) with pre-coated silica gel plates from EMD Chemicals (TLC Silica gel 60 F254, 250 $\mu$m thickness) using UV light as the visualizing agent and an acidic mixture of anisaldehyde, phosphomolybdic acid (PMA), chromic acid, iodine vapor, Seebach's stain, or basic aqueous potassium permanganate (KMnO$_4$), and heat as developing agents. Preparatory thin layer chromatography (PTLC) was performed using the aforementioned silica gel plates. Flash column chromatography was performed over silica gel 60 (particle size 0.035- 0.07 mm) from Acros Organics. NMR spectra were recorded on Bruker DRX-600 (equipped with a 5 mm DCH Cryoprobe), AV-600, DRX-500 or DPX-400 and calibrated using residual non-deuterated solvent as an internal reference (CHCl$_3$ @ 7.26 ppm $^1$H NMR, 77.16 ppm $^{13}$C NMR; (CD$_3$)$_2$CO @ 2.05 $^1$H NMR, 206.26 $^{13}$C NMR; CD$_3$OD @ 3.31 $^1$H NMR, 49.00 $^{13}$C NMR). The following abbreviations (or combinations thereof) were used to explain the multiplicities: s = singlet, d = doublet, t = triplet, q = quartet, p = pentet, sex = sextet, sep = septet m = multiplet, br = broad. LC/MS analysis was performed on an Agilent 1200 series HPLC/MS equipped with an Agilent SB-C18 2.1 mm × 50 mm column, with mass spectra recorded on a 6120 Quadrupole mass spectrometer (API-ES), using MeCN and H$_2$O as the mobile phase (0.1% formic acid). LC-MS runs used the following method unless otherwise specified: flow rate of 0.5 mL/min is used, initial equilibration of 5% MeCN/H$_2$O with a linear

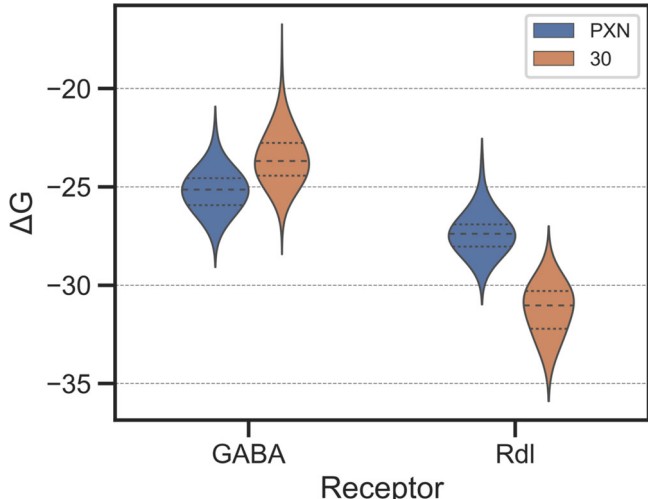

**Fig. 6 | Distributions of in silico binding free energies of PXN (1) and 30 to α1β3γ2L GABA$_A$ and RDL.** Computed binding free energies predict that **30** is not as potent as PXN (**1**) toward the GABA$_A$ receptor (ΔΔG = +1.8 ± 1.5), whereas **30** is much more potent than PXN (**1**) toward the RDL receptor (ΔΔG = −3.9 ± 1.6), in agreement with experimental data. A standard, two-tailed *T*-test showed statistical significance in the differences between the binding free energies of PXN (**1**) and **30** for the GABA$_A$ receptor ($p = 2.95×10^{-30}$) and the RDL receptor ($p = 1.22×10^{-31}$). Each distribution except for **30**@GABA$_A$ passed at least one test of normality (Shapiro–Wilk, D'Agostino's K2, Anderson-Darling). Small dotted lines indicate the interquartile range of predicted binding free energies, while the thick dotted line indicates the 50th percentile.

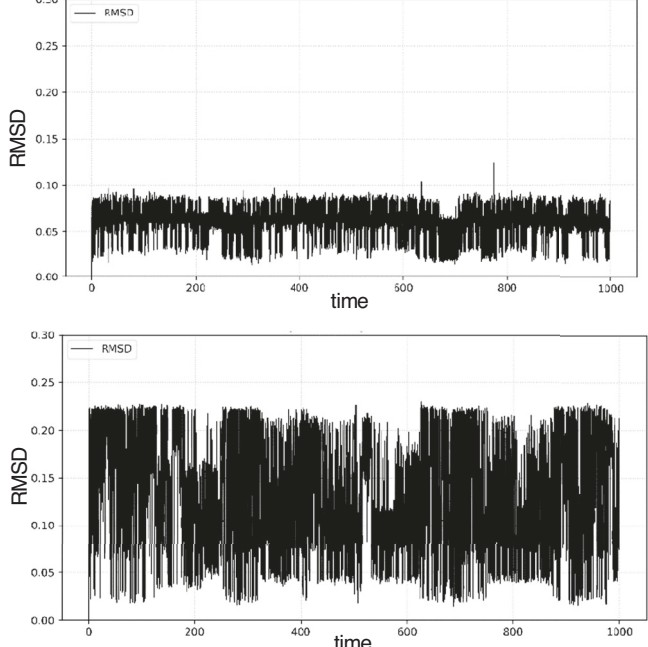

**Fig. 7 | Molecular dynamics comparisons.** Overall horizontal plots of PXN (**1**) and **30** suggest neither ligand leaves its binding site in the RDL receptor, despite the voluminous channel.

gradient to 95% MeCN / H$_2$O over 5 min, then a hold at 95% MeCN / H$_2$O for an additional 3 min. Optical rotations were measured digitally on an Autopol III polarimeter from Rudolph Research Analytic, using a flow cell with a 0.5 decimeter pathlength and the sodium lamp D-line wavelength (λ = 589.3 nm). High-resolution mass spectrometric data

were obtained on a Waters Xevo G2-XS QTOF instrument. Commercial picrotoxinin was obtained from Millipore Sigma (USA) and used for relative analog preparation.

### Reporting summary

Further information on research design is available in the Nature Portfolio Reporting Summary linked to this article.

## Data availability

The data supporting the findings of this study are available within the paper and its Supplementary Information. Details about materials and methods, experimental procedures, characterization data, and NMR spectra are available in the Supplementary Information. All other data are available from the corresponding author upon request. The X-ray crystallographic coordinates for structures reported in this study have been deposited at the Cambridge Crystallographic Data Center (CCDC) under deposition numbers CCDC2048201 (**11**), CCDC2081332 (**13**), CCDC2071501 (**14**), CCDC2048202 (**20**), CCDC2048203 (**21**), CCDC2069870 (**23**), CCDC2079783 (**30**), CCDC2078168 (**31**), and CCDC2087432 (d$_9$-**34**). These data can be obtained free of charge from The Cambridge Crystallographic Data Centre via www.ccdc.cam.ac.uk/data_request/cif. Calculated coordinates of the optimized structures are available in the Source Data file. Source data are provided with this paper.

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

## Acknowledgements

This work was financially supported by National Institutes of Health (R35 GM122606) and JITRI (Fellowship to G.T.). Dr. Milan Gembicky, Dr. Erika Samolova, Dr. Jake Bailey, Professor Arnold Rheingold and the UCSD Crystallography Facility are acknowledged for X-ray crystallographic analysis. Dr. Laura Pasternack and Dr. Dee-Hua Huang are acknowledged for assistance with NMR spectroscopy.

## Author contributions

R.A.S., G.T., J.W., and Z.B. conceived the project. R.A.S. directed the research and composed the manuscript, and G.T. compiled the Supplementary Information section. G.T. executed all synthetic and spectroscopic work, S.G. and K.B. undertook electrophysiology on RDL, S.C. and A.C. contributed to the computational analyses of reaction

coordinates in Figs. 3 and 4, and A.S. contributed the computational binding studies in Fig. 5.

## Competing interests

A patent related to the synthesis of PXN has been filed: Crossley, S. W. M.; Tong, G.; Lambrecht, M. J.; Burdge, H.; Shenvi, R. A. "Synthesis of picrotoxinin and related compounds" U.S. Patent Application No. 16/947,560, Filing Date: 08/06/2020; Scripps Ref No 2036.0 / TSR2454P. The remaining authors declare no competing interests.
