## [Peer Review File · Nature Communications]

REVIEWER COMMENTS

Reviewer #1 (Remarks to the Author):

This paper by Shenvi, Chen and Corteva is excellent. It's a tour de force of synthesis, merged with computation and will serve as an excellent modern example of the role of synthesis in agrobusiness, which is highly relevant. Both agrochemicals and pharma are highly focused on synthetic accessibility and the role of computation in "morphing" lead compounds....usually there are significant limitations to such exercises due to a lack of synthetic bravery. Here, the team has tackled some highly complicated targets, informed by computation but accessed by experimental total synthesis.

The work describes the synthesis of methylated variants of picrotoxinin, and various other picrotoxinin analogs, that are tested for activity against GABA_A receptors. The first synthesis of picrotoxinin from this team was earlier described, and indeed the current synthesis is of the highest quality. The methylated analogs have improved chemical and metabolic stability and selectivity for targeting invertebrates over mammalian GABA_A. I have no concerns about the science, the supporting information accurately supports the conclusions of the work. This paper is highly exciting to me, and showcases the state of the art of merging computation with synthesis. There has long been calls from the pharmaceutical and agrochemical industries to return to natural products, but synthetic accessibility has limited the ability to do so. This latest work from Shenvi et al showcases how it can be done. I think both industrial and academic chemists will benefit greatly from the work.

Reviewer #2 (Remarks to the Author):

The manuscript entitled "C5 Methylation Confers Accessibility, Stability and Selectivity to Picrotoxinin" by Shenvi, Chen, and their co-authors describes comprehensive computational, chemical, and biological studies on a series of C5 methylated picrotoxinin compounds. The study is noteworthy for a number of reasons:

1. Usually both total synthesis and semi-synthesis approaches often aim at simplifying molecular structure while retaining biological activity (and preferably improving other parameters, "reduce to the max"). This study provides detailed evidence for the conundrum that more complex structures (according to the Böttcher scale) can lead to improved properties. In particular, careful molecular editing by the authors through synthetic chemistry led to profound insight into the biological consequences.

2. From a biological perspective, the main finding of this study that selectivity for invertebrate receptors can be increased via total synthesis is very interesting, and has ramifications to other fields beyond chemistry and chemical biology. In fact, selective targeting of invertebrate receptors could lead to translation into crop protection applications.

3. From a mechanistic perspective, the main finding that the additional methyl group does not induce kinetic selectivity via Bürgi-Dunitz trajectory, but instead via conformational reorganisation and downstream epoxide opening is truly fascinating. It illuminates the intricate structural requirements leading to undesired pathways, and by the careful analyses of the authors, their deep-rooted knowledge into chemical structure and reactivity, and consequently the use of total synthesis to evaluate these hypotheses is truly remarkable to the field.

For these reasons, the manuscript definitely advances the field in that total synthesis, and the addition of complexity, can lead to profound mechanistic insight, leading to compounds with improved selectivity and translational potential in crop science. Therefore, it is perfectly suited for the broad scientific audience of the journal.

The claims are fully supported by experimental evidence, and the hypotheses formulated are appropriately addressed and evaluated. The work is original, and while it builds on a previous study by the authors, the main findings are truly novel and much beyond of the scope to what has been previously reported.

The work is reported to the high standards as required in the field, and the methods and SI sections allow for full reproducibility of this study.

Overall, there are only a number of small corrections to be addressed by the authors, which are listed below. The manuscript is recommended for acceptance.

- Abstract: It should read "We observe a remarkable stabilizing effect of a C5 methyl group" instead of "We observe a remarkable stabilizing effect of a C5 methyl".

- Compounds should be numbered according to their appearance in the main text.

- Adams' catalyst not Adam's.

- Legend Figure 2: "Figure 2. Twenty-one analogs prepared for biological " not "Figure 2. Twenty-one analogs prepared for biological "
- "We wondered whether replacement of one or both C12 and C6 hydroxyls" hydroxyl refers to the radical species, shouldn't this be hydroxy groups? There is another use of hydroxy later in the manuscript.
- The fitted lines for PXN (1) in Figure 3 should cross the origin (0/0). Otherwise, it does not make any physical (or physiological) sense
- Figure 3: The term 'right lactone' is misleading. Use C numbering instead.

Reviewer #3 (Remarks to the Author):

The authors describe the synthesis of a novel set of analogs enabled by the route development reported previously by this group (JACS 2020). The introduction of a methyl group was shown to lead to synthetic simplification of the target and decreased metabolic decomposition leading to improved activity. I support publication after the minor points below are addressed.

The carbocation intermediate is quite surprising for the conversion of 13 to 30, and the ability of DAST to lead to the tertiary fluoride is also striking. Have activated tertiary alcohols previously been converted to the corresponding fluorides? If so, citation to these references would resolve the concern.

Is it possible that the base mediated conditions trigger a retro-aldol, followed by fluoride attack of the ketone and re-closure of the ring could lead to the same product? It would not account for the formation of 31, but 30 may be formed by a mechanistically distinct pathway.

The comparison in reactivity between the methyl analog of PXN is remarkable and underscores the value of this general research direction. It is a profound difference in reactivity that is observed under mildly acidic or basic conditions and a more modest difference is observed by esterase-mediated hydrolysis (2-3x). Nonetheless, this modification shows that introduction of substituents to C5 leads to decreased rates of decomposition.

The computed pathway in Figure 3g refers to the basic conditions of hydrolysis prompted by NaOMe, but the authors have selected to model this by the nucleophilic addition of MeOH. Could

the authors add some discussion to the text about this simplification? Presumably this change is made to avoid the complicating factors of studying the anionic additions that are complicated by counterion and differences due to solvation. I think this is a reasonable simplification that should be addressed in the text.

The authors comment on the presence of a steric interaction between the methyl group and the nearby hydrogen atoms with distances noted in Figure 4. Do the authors have evidence that this is a repulsive interaction? I would expect these interactions to be attractive dispersive interactions.

How did the authors select the level of theory for the methanolysis? The authors should cite a related study that employed this level of theory or conduct the appropriate benchmarking study.

It's not clear that the docking study has identified the correct mode of binding, but the hypothesis included here accounts for several of the experimental observations. This binding model and the analysis thereof should have value to future researchers in this area and can be revised in the future as needed.

Point-by-Point Response to Referees

We thank the reviewers for their insightful suggestions and comments. We have addressed the following items in this point-by-point response and made changes to the manuscript as noted.

Reviewer #1

This paper by Shenvi, Chen and Corteva is excellent. It's a tour de force of synthesis, merged with computation and will serve as an excellent modern example of the role of synthesis in agrobusiness, which is highly relevant. Both agrochemicals and pharma are highly focused on synthetic accessibility and the role of computation in "morphing" lead compounds....usually there are significant limitations to such exercises due to a lack of synthetic bravery. Here, the team has tackled some highly complicated targets, informed by computation but accessed by experimental total synthesis.

The work describes the synthesis of methylated variants of picrotoxinin, and various other picrotoxinin analogs, that are tested for activity against GABA_A receptors. The first synthesis of picrotoxinin from this team was earlier described, and indeed the current synthesis is of the highest quality. The methylated analogs have improved chemical and metabolic stability and selectivity for targeting invertebrates over mammalian GABA_A. I have no concerns about the science, the supporting information accurately supports the conclusions of the work. This paper is highly exciting to me, and showcases the state of the art of merging computation with synthesis. There has long been calls from the pharmaceutical and agrochemical industries to return to natural products, but synthetic accessibility has limited the ability to do so. This latest work from Shenvi et al showcases how it can be done. I think both industrial and academic chemists will benefit greatly from the work.

Response: We appreciate the reviewer's positive comments. No action items were provided.

Reviewer #2

The manuscript entitled "C5 Methylation Confers Accessibility, Stability and Selectivity to Picrotoxinin" by Shenvi, Chen, and their co-authors describes comprehensive computational, chemical, and biological studies on a series of C5 methylated picrotoxinin compounds. The study is noteworthy for a number of reasons:

1. Usually both total synthesis and semi-synthesis approaches often aim at simplifying molecular structure while retaining biological activity (and preferably improving other parameters, "reduce to the max"). This study provides detailed evidence for the conundrum that more complex structures (according to the Böttcher scale) can lead to improved properties. In particular, careful molecular editing by the authors through synthetic chemistry led to profound insight into the biological consequences.
2. From a biological perspective, the main finding of this study that selectivity for invertebrate

receptors can be increased via total synthesis is very interesting, and has ramifications to other fields beyond chemistry and chemical biology. In fact, selective targeting of invertebrate receptors could lead to translation into crop protection applications.

3. From a mechanistic perspective, the main finding that the additional methyl group does not induce kinetic selectivity via Bürgi-Dunitz trajectory, but instead via conformational reorganisation and downstream epoxide opening is truly fascinating. It illuminates the intricate structural requirements leading to undesired pathways, and by the careful analyses of the authors, their deep-rooted knowledge into chemical structure and reactivity, and consequently the use of total synthesis to evaluate these hypotheses is truly remarkable to the field.

For these reasons, the manuscript definitely advances the field in that total synthesis, and the addition of complexity, can lead to profound mechanistic insight, leading to compounds with improved selectivity and translational potential in crop science. Therefore, it is perfectly suited for the broad scientific audience of the journal.

The claims are fully supported by experimental evidence, and the hypotheses formulated are appropriately addressed and evaluated. The work is original, and while it builds on a previous study by the authors, the main findings are truly novel and much beyond of the scope to what has been previously reported.

The work is reported to the high standards as required in the field, and the methods and SI sections allow for full reproducibility of this study.

Overall, there are only a number of small corrections to be addressed by the authors, which are listed below. The manuscript is recommended for acceptance.

Response: We are grateful for this reviewer's positive comments and the level of detail they provided. We have made these changes in the revised manuscript as recommended.

- Abstract: It should read "We observe a remarkable stabilizing effect of a C5 methyl group" instead of "We observe a remarkable stabilizing effect of a C5 methyl".

Response: Thanks for the comments. We have revised "C5 methyl group" to "C5 methyl".

- Compounds should be numbered according to their appearance in the main text.

Response: We have mostly followed ordered numbering, except in cases where it strains the text. In addition, we have extensively revised the naming of compounds in the text to unify the figures and clarify the discussion for readers.

- Adams' catalyst not Adam's.

Response: Thanks for the comments. We have revised "Adam's catalyst" to "Adams' catalyst".

- Legend Figure 2: "Figure 2. Twenty-one analogs prepared for biological " not "Figure 2. Twenty-one analogs prepared for biological "

Response: Thank you for catching this error; it has been corrected.

- "We wondered whether replacement of one or both C12 and C6 hydroxyls" hydroxyl refers to the radical species, shouldn't this be hydroxy groups? There is another use of hydroxy later in the manuscript.

Response: Thanks for the comments. "C12 and C6 hydroxyls" has been revised to "C12 and C6 hydroxy groups".

- The fitted lines for PXN (1) in Figure 3 should cross the origin (0/0). Otherwise, it does not make any physical (or physiological) sense.

Response: This is an important point and we're grateful to the referee for their attention to detail. We have removed the trendline for clarity so the readers can evaluate the data unambiguously.

It's worth noting that the prior line accounted for experimental errors in accurate detection of concentration because the concentration of analyte would not have increased by the second time point; the trendline represented extrapolation based on the more reasonable data points at later times. We could also retain the trendline and plot the data against the analyte peak area: same data, same trend, but it would not appear to violate physics!

- Figure 3: The term 'right lactone' is misleading. Use C numbering instead.

Response: That is a good suggestion. "right lactone" has been changed to "C15 lactone".

Reviewer #3

The authors describe the synthesis of a novel set of analogs enabled by the route development reported previously by this group (JACS 2020). The introduction of a methyl group was shown to lead to synthetic simplification of the target and decreased metabolic decomposition leading to improved activity. I support publication after the minor points below are addressed.

Response: We thank the reviewer for the excellent comments. The following is our point-by-point response to the specific comments.

The carbocation intermediate is quite surprising for the conversion of 13 to 30, and the ability of DAST to lead to the tertiary fluoride is also striking. Have activated tertiary alcohols previously been converted to the corresponding fluorides? If so, citation to these references would resolve the concern.

Response: We thank the reviewer for this comment. The answer is yes: tertiary alcohols have been converted to their fluorides in the past. We have included a helpful reference for the interested reader. Reference 37 has been cited in the revised manuscript.

37. Middleton, W. J. New fluorinating agents: Dialkylaminosulphur fluorides, *J. Org. Chem.* **40**, 574-578 (1974).

Is it possible that the base mediated conditions trigger a retro-aldol, followed by fluoride attack of the ketone and re-closure of the ring could lead to the same product? It would not account for the formation of 31, but 30 may be formed by a mechanistically distinct pathway.

Response: This is an interesting point. The pathway is illustrated below. It's possible that this mechanism is operative to form 30. In our view, the surprising result is formation of 31, which we

would prefer to illustrate in Figure 3. If the referee would prefer, we could revise the SI to include this pathway. A common pathway involves alcohol ionization and cation capture, which is commonly induced by DAST in, for example, conversion of 1-adamantanol to 1-fluoro-adamantane (see also Volkman *et al.* *JACS* **2000** *122*, 466).

The computed pathway in Figure 3g refers to the basic conditions of hydrolysis prompted by NaOMe, but the authors have selected to model this by the nucleophilic addition of MeOH. Could the authors add some discussion to the text about this simplification? Presumably this change is made to avoid the complicating factors of studying the anionic additions that are complicated by counterion and differences due to solvation. I think this is a reasonable simplification that should be addressed in the text.

Response: We thank the reviewer for this comment. To clarify, Figures 3f and 3g both model base-catalyzed methanolysis pathways, not hydrolysis pathways. These calculations were performed to rationalize experimental results shown in Figure 3a–d, where NaOMe-catalyzed methanolysis carried out in d_4 -methanol revealed favored pathways (C15 nucleophilic attack followed by intramolecular epoxide opening) with a methoxide nucleophile.

Our computational model therefore used the methoxide (MeO^-) as the nucleophile to reflect the basic reaction conditions used. In **TS-1** (Figure 1f), for example, the nucleophile attacking the carbonyl group is a methoxide. Because the resultant anionic intermediates (such as **B** and **F**) eventually undergo protonation upon acidic workup, we described the transformation as a nucleophilic addition of MeOH in the energy diagram captions. We hope this clarification appropriately addresses the reviewer's concern.

The authors comment on the presence of a steric interaction between the methyl group and the nearby hydrogen atoms with distances noted in Figure 4. Do the authors have evidence that this is a repulsive interaction? I would expect these interactions to be attractive dispersive interactions.

Response: We thank the reviewer for this comment. Typically, attractive (stabilizing) dispersive interactions between alkyl groups require H...H distances to fall within the 2.5–3.0 Å range (*Angew. Chem. Int. Ed.* **2016**, *55*, 8086; *J. Am. Chem. Soc.* **2017**, *139*, 16548). With rare exceptions observed in certain polyhedrane dimers (*Nat. Chem.* **2011**, *3*, 323), H...H distances smaller than 2.4 Å (the sum of the van der Waals radii of two H atoms) are seldom attractive or stabilizing in nature, as steric repulsion increases dramatically at this range. As all H...H distances labeled in Figure 4 are in the 2.0–2.4 Å range, they are most likely repulsive interactions.

How did the authors select the level of theory for the methanolysis? The authors should cite a related study that employed this level of theory or conduct the appropriate benchmarking study.

Response: We thank the reviewer for this comment. We have included references to studies that successfully employ this level of theory in the revised manuscript to provide support for our choice of computational methods.

"...ωB97X-D/def2-TZVPPD, SMD (H₂O)// ωB97X-D/def2-SVP, SMD (H₂O) level of theory in Gaussian 16, which has been shown to perform well in the modeling of transformations involving anionic intermediates.^{48,49}"

48. King, C. R., Holdaway, A., Durrant, G., Wheeler, J., Suaava, L., Konnick, M. M., Periana, R. A. & Ess, D. H. Supermetal: SbF₅-mediated methane oxidation occurs by C–H activation and isobutane oxidation occurs by hydride transfer. *Dalton Trans.* **48**, 17029–17036 (2019).

49. Klunda, T., Hricovíni, M., Šesták, S., Kóňa, J. & Poláková, M. *New J. Chem.* **45**, 10940–10951 (2021).

REVIEWERS' COMMENTS

Reviewer #3 (Remarks to the Author):

All concerns have been addressed. I fully support publication of this excellent study!